# Advanced Strategic Research to Promote the Use of Rice Genetic Resources

**Jae-Sung Lee** [1] , **Dmytro Chebotarov** [1] , **John Damien Platten** [2] , **Kenneth McNally** [1,*] and **Ajay Kohli** [1]

1    Strategic Innovation Platform, International Rice Research Institute, Los Baños, College, Laguna 4031, Philippines; js.lee@irri.org (J.-S.L.); d.chebotarov@irri.org (D.C.); A.Kohli@irri.org (A.K.)
2    Rice Breeding Platform, International Rice Research Institute, Los Baños, College, Laguna 4031, Philippines; j.platten@irri.org
*    Correspondence: K.McNally@irri.org

**Abstract:** International genebanks have a collection of over 760 K conserved accessions of various plants, most of these accessions are within the multi-lateral system governed by the International Treaty on Plant Genetic Resources for Food and Agriculture (ITPGRFA). However, in spite of the success in collection and conservation, only a small portion of the genetic diversity has been used in crop breeding programs. As climate change-induced new or enhanced constraints seriously hamper crop productions, researchers and breeders should be able to swiftly choose an appropriate set of genetic resources from the genebank and use them for improving crop varieties. Here, we present some advanced technologies that can effectively promote the use of diverse rice accessions held at national/international genebanks. High throughput phenotyping using multispectral imaging systems and unmanned aerial vehicles (UAV) can quickly screen large numbers of accessions for various useful traits. Such data, when combined with that from the digital rice genebank consisting of genome sequencing data, will significantly increase the efficiency in breeding efforts. Recent genome sequencing data of the rice wild species will also add to the resources available for pre-breeding efforts such as the introgression of useful genes into modern rice varieties. We expect that these advanced technologies and strategies developed through the global rice research programs will be applicable for many closely related species as well.

**Keywords:** plant genetic resources; rice; utilization; climate change; strategic research; digital genebank; breeding

## 1. Introduction

Plant genetic resources (PGR) held in national/international genebanks are a hope for the future of global food security. Uncharacterized accessions in a diverse germplasm collection may be the source of traits sought by researchers and breeders to swiftly respond to new types or enhanced levels of biotic and/or abiotic stresses induced by climate change. At this time, the International Treaty on Plant Genetic Resources for Food and Agriculture (ITPGRFA) provides the mechanism by which international genebanks of the Consultative Group on International Agricultural Research (CGIAR), a consortium of international agricultural research centers, share the extensive germplasm collection and effective ex situ conservation for 760,467 accessions (as of 2019) for a range of plants [1] amassed over the years. These fed the crop breeding programs that led the green revolution in many developing countries. For example, the International Rice Genebank (IRG) of the International Rice Research Institute (IRRI) conserves the World Rice Collection composed of 140 K diverse rice accessions, breeding material, and elite varieties. Over 120 countries donated these accessions to be conserved and

held for humanity. By harnessing such genetic diversity, IRRI and partner countries have developed outstanding rice varieties starting with IR8, a high-yielding semi-dwarf variety called "miracle rice" that avoided potential famine situations [2].

In spite of past successes, the need for research on characterizing and utilizing genetic resources remains high due to serious challenges of climate change and not least due to the predicted demographic, land-use, and socio-economic factors [3]. Rice adaptation to climate change and other imperatives will need to keep pace with the rate of change. However, current breeding cycles cannot respond to novel constraints on crop production that assume stability of other critical parameters. For example, there remains a grave paradox of elevated $CO_2$ increasing the efficiency of photosynthesis, potentially leading to increased yield, but negatively affecting the nutritional quality of the staple grains, tubers, and also vegetables [4]. Moreover, the concomitant high temperature is expected to lead to reduced yield by unfavorable effects on pollen development and sink starch metabolism [5]. High temperature also accelerates the growth rate of crop damaging insect pests such as yellow stem borer [6]. Increased emergence of such pest challenges, quickly becoming critical, can happen in much shorter timelines than the variety breeding cycles. However, Aggarwal et al. (2019) indicate that for some crops, such as maize and wheat, changing crop management approaches may be sufficient to correct for climate change induced yield losses [7]. Similarly, the altered frequencies of weather events, especially typhoons, at late seed maturity stage can cause conditions that are favorable for damage to the rice crop through pre-harvest sprouting (PHS) [8,9]. Muehe et al. [10] reported that during flooding, the availability of soil arsenic to plants increases, causing 39% reduction in rice grain yield. The frequency and severity of such floods has become increasingly unpredictable due to climate change. Such challenges demand scientific innovations promoting fast, accurate, and cost-efficient breeding pipelines to generate resilient varieties as illustrated by Atlin et al. [11].

Rice is a staple crop for more than half of the world population, and it is also rich in genetic resources [12,13]. Furthermore, rice has relatively small genome size and close relationship to other major grain crops. These characteristics led to its genome being the first to be sequenced crop genome [14]. Therefore, the principles developed through advanced rice research could be a good model for other crop species, particularly inbred species that have ex situ collections [12]. This opinion paper aims to a) discuss advanced strategic research bridging the gap between genebank collections and breeding products, and b) recommend the future direction and practical ways to promote the use of rice genetic resources for global rice breeding programs.

## 2. Expansion of Ex Situ Conservation Research

Genetic erosion, the loss of alleles or genes from the genetic diversity, may occur when genetic drift results in high frequencies of non-adaptive gene variants and also under natural selection in conditions with significant changes in plant growth environment. Modern breeding programs highly focused on a few important traits of the day can also contribute to genetic erosion [15]. Genebanks hold ex situ collections to reduce such genetic erosion. Conservation research is a key activity to predict, monitor, and manage seed longevity of the ex situ collections. Conservation research can also benefit farmers in terms of seed security. In most cases in developing countries, farmer's seed lots are stored at ambient conditions, which leave seed viability and vigor vulnerable to the increased frequency of hot and humid weather under climate change [16]. Lower seed vigor can result in poor crop establishment in direct-seeding cultivation, and even more severely under unfavorable crop environments such as drought [17,18]. While determining seed longevity of genebank accessions, donor germplasm having high seed longevity could be selected and used for the breeding work enhancing seed longevity of popular crop varieties [19]. Sasaki et al. [20] reported that *qLG-9*, the quantitative trait locus (QTL) conferring pronounced seed longevity, was successfully introgressed into Nipponbare, one of the most popular temperate japonica rice varieties in Japan. Further efforts to enhance seed longevity of other elite rice varieties are needed in the near future [16].

Another aspect of expanding conservation research for breeding applications is that seed dormancy is a critical trait for both management of crop wild relatives held at genebank and development of climate-resilient crop varieties. Genebank managers often experience difficulties in seed germination of wild relatives having a strong dormancy behavior as part of their strategy for survival in nature [21]. To ensure successful regeneration of crop wild relatives, intensive biological research is required for better understanding of seed dormancy mechanisms as well as development of species-specific dormancy breaking protocols [22,23]. In contrast, modern crop varieties tend to have a weak dormancy caused by the domestication of wild ancestors [24]. Under climate change impacts, reduced dormancy can result in serious PHS damage on economically important crops such as wheat and rice [8,25]. Upstream research addressing physiological and genetic aspects of seed dormancy could be a part of solution for better management of wild relatives as well as improving modern crop varieties.

## 3. Pre-Screening for Accelerating PGR Research

The genetic diversity conserved in genebanks contains useful alleles that could confer desirable traits, including those that would enable adaptation to rapid climate change as well as to paradigm shifts in the value in crops, e.g., consumption for health promoting properties [26,27]. Genebanks are in a good position to observe various plant/seed phenotypes during routine operations such as regeneration of seed or monitoring seed viability. Some traits underlying these phenotypes are unique and potentially important for the future breeding perspectives. For example, in the crop–livestock research area, high plant biomass is a desirable trait to improve the productivity of cereal crops for forage use [28,29]. In the case of rice, the National Institute of Crop Science (NICS) of South Korea introduced IRRI genebank germplasm and developed high biomass yielding varieties with multiple disease and insect resistances [30]. For genebank managers, it is not difficult to find the accessions having an outstanding biomass during the regeneration and field-characterization of a diverse crop panel. The stay-green (SG) is another promising trait that is easily scored in the regeneration field. SG maintains a green leaf color with high photosynthesis capacity until late grain-filling stage, especially under drought and heat stress conditions and secures grain yield [31]. Therefore, this trait is potentially important in future breeding programs to develop climate-smart crop varieties [32]. Although SG is most informative in such stress situations, we have observed that ambient temperature at late vegetative and seed maturity stages rose during the last few years and there were large variations in SG, which allowed us to select a few accessions having high SG.

Multispectral imaging, through the use of an instrument such as the Videometer, is a fast and robust approach helpful for managing genebank accessions [33,34]. This non-destructive technique can measure various seed phenotypes, e.g., seed size, shape, and color within a few seconds (Figure 1a). It can also perform seed purity tests to compare regenerated seeds with the original seeds [33]. Sharing basic information of seed phenotypes through the genebank website would be helpful for seed users to choose the right accessions based on their purposes. Multispectral imaging, such as that obtained by the Videometer, is very useful in healthier crops research programs too. Red, purple or any other colored pericarps in seed contain health promoting phenolic compounds [35,36]. Comparing with conventional biochemical assays that require expensive operational and labor costs as well as destruction of many seeds, multispectral imaging analysis can provide reliable data of color-based traits with minimal cost [37]. Some advanced institutes are equipped with automated-phenotyping systems to assess key agronomic traits such as seedling vigor in controlled environments [38]. CGIAR-genebanks also use automation systems such as Germination Scanalyzer to improve the accuracy and efficiency of seed viability monitoring for seed lots newly harvested or stored in cold rooms (Figure 1b). While counting total number of germinated seeds as well as measuring the uniformity of germination (indicating seed vigor), the image analysis software can measure additional seed/seedling traits, e.g., primary root length and area of secondary/tertiary roots, which are very useful information for seed users. Under drought conditions, tolerant plants tend to produce deep roots at early seedling stage, which increases the efficiency of water and nutrient uptake [39]. Hence, genebank germplasm

showing a long primary root during the germination test have merit for further drought screening. On the other hand, in phosphorus- or zinc-deficient soils, rapid development of crown roots and shallow rooting concentrated in nutrient-rich regions are key tolerant mechanisms [40,41]. Therefore, genebank accessions having high secondary/tertiary root density could be used for research on nutrient deficiency tolerance in crops.

Unmanned aerial vehicle (UAV)-based high throughput phenotyping is the most promising way to screen a large number of accessions, efficiently and effectively (Figure 1c). A single operator with a UAV and aerial image analysis software can cover the whole field site. In 10 min for UAV operation per hectare plus 1 h for image analysis, characterization of various agronomic traits such as leaf color, plant height, biomass, and flowering time can be obtained [42,43]. This new technology can significantly reduce labor and costs as compared with manual phenotyping methods requiring many genebank operators' time and effort. In addition to plant characterization, the aerial imagery provides useful information for field-risk management. For example, pale leaf color (indicating nitrogen deficiency), leaf rolling (as drought responses), weed growth, disease infection, and lodging [44,45] can be monitored and the necessary actions undertaken. Furthermore, high performance cameras capturing multispectral and thermal imagery can determine the stress tolerance-indicators under high temperature, drought, or salinity conditions (Figure 1c). As part of the capacity building, IRRI has developed a manual on UAV-phenotyping and has also held training courses for partner countries. Yang et al. [46] provide a comprehensive review of high throughput phenomics from controlled environment to field-based systems; all of these approaches will benefit the characterization of diverse genetic resources.

The above-mentioned high throughput methods are maturing technologies and promise to greatly extend the scope of phenotyping of genebank germplasm. However, data analysis for image and other data obtained from high-throughput phenotyping requires high-performance computing and/or cloud-based solutions. Application of novel machine learning (ML) and AI based approaches will facilitate extraction of features and parameters that can serve as new traits or proxies for measurements classically obtained by hand. Combined with increase in genomic data, as we will argue below, this requirement necessitates a dedicated module for data storage towards a modern, digital genebank.

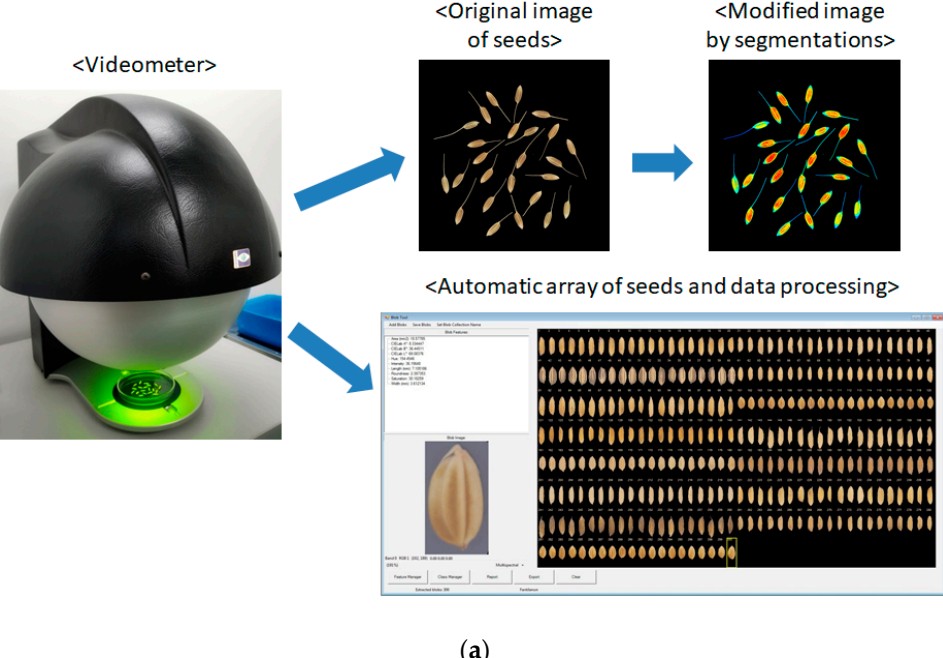

(**a**)

**Figure 1.** *Cont.*

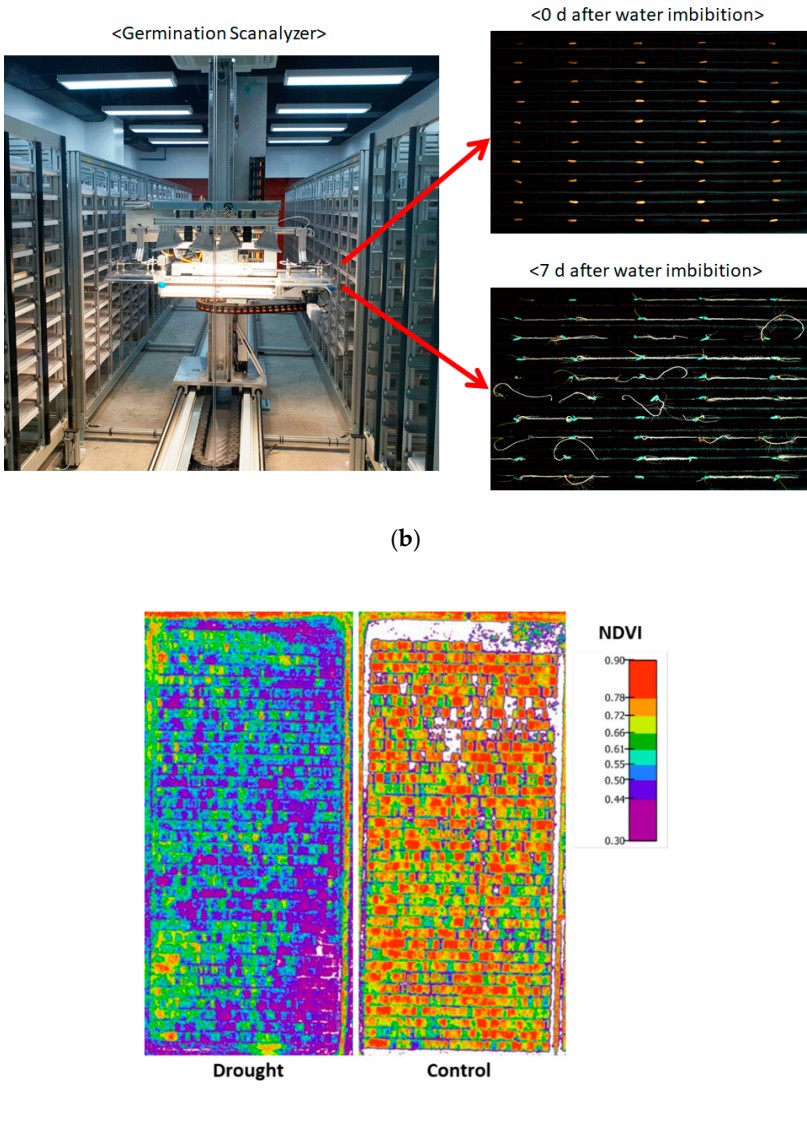

**Figure 1.** Automated-phenotyping system for screening the rice collection held at the International Rice Research Institute (IRRI) genebank. (**a**) Videometer (Videometer A/S, Herlev, Denmark) for performing multispectral imaging analysis of seed samples. In the modified image by segmentations, red and yellow parts are the areas of interest, i.e., measuring seed size, shape, and color, whereas blue regions, such as awn and glume, are less important depending on the objective; (**b**) Germination Scanalyzer (LemnaTec GmbH, Aachen, Germany), the automated-phenotyping system for seed viability monitoring. The system is housed in a semi-climate controlled room that can accommodate 720 seed-trays at the same time and analyze various traits such as number of germinated seeds, uniformity of germination, primary root length, and area of secondary/tertiary roots; (**c**) Unmanned aerial vehicle (UAV)-thermal imagery for drought tolerance screening in the rice paddy. Accessions are planted in the same positions in both sides of the field. NDVI: Normalized difference vegetation index. The index scale at 0.90 and 0.30 indicate completely healthy and unhealthy plants, respectively.

## 4. Digital Rice Genebank for Game-Changing Research

Genebanks have an innate obligation to make the germplasm collection and the accompanying identification information available for distribution to users. Strategies promoting the use of genebank accessions may compensate the high cost of germplasm collection and conservation. Although the genetic diversity conserved at genebanks contains most of the resources to address current and

potentially the future-constraints in crop production, it requires researchers extensive screening of the germplasm collection for various traits in multiple trials, which is inefficient and simply not possible. One promising approach to reduce the time and labor for choosing the most appropriate set of accessions for specific research goals is to provide the seed users with the genotype/sequencing data [47]. Based on allelic information in the specific region of interest, e.g., the *Sub1* locus on chromosome 9 in rice conferring submergence tolerance [48], the users can narrow the list of germplasm they require for their research, thus significantly reducing the costs for evaluation and accelerating the discovery process.

Advanced genotyping methods such as next generation sequencing (NGS) have evolved through human genome projects [49,50] and become applicable for crop genetics research. Among crop species, rice research benefited greatly from such sequencing technologies and took the lead in crop informatics with the support of international rice community including the CGIAR system and donor countries [12]. McCouch et al. [51] provided the high-density rice array (HDRA) comprised of 700,000 single-nucleotide polymorphisms (SNPs) from the 1593 rice accessions held at genebanks. This first data set was highly efficient to strengthen the genetic research for genome-wide association studies (GWAS), enabling researchers to quickly discover novel loci/genes associated with desired traits such as grain size, cooking quality, root cone angle, salinity, and Zn-deficiency tolerances [12,41,52–54]. Once the identified loci are validated, researchers can easily choose the right genebank materials based on the presence of particular genetic markers associated with target traits. With the financial support from the Bill and Melinda Gates Foundation and the Chinese Ministry of Science and Technology, the 3000 Rice Genomes Project sequenced 3010 diverse rice accessions representing 15 subgroups derived from *indica, aus,* and *japonica* major groups and discovered over 18 million SNPs [55,56]. All genotype data with supportive phenotype data generated through HDRA and 3K projects is publicly available so that seed users worldwide can request germplasm from the IRG, based on sequencing information [57,58]. As of 31 May 2020, 768 studies cited either SNP-Seek or HDRA-related work, according to Scopus [59]. This work was largely possible due to availability of multiple software tools developed within global bioinformatics community. Variant discovery is enabled by advances in algorithms and their efficient implementations in software like BWA [60], SAMtools [61], Picard toolkit [62], Genome Analysis Toolkit [63,64], and others. Continued support of tool development and adjustment is needed for further developments in sequence variation discovery.

In spite of these successful achievements, it has become increasingly clear that for NGS data of large populations, the standard variant discovery approaches cannot capture all useful variation when only one reference genome is used. Inevitably, large and potentially useful population- and individual-specific genomic regions are left unexplored if one uses only one reference genome [56,65–70]. When using a reference genome specific to an ecotype, a more complete picture of variation is captured, including the variants missing in a single reference genome. For rice, reference genomes from three popular varieties: Minghui 63, Zhenshan 97B, and Nipponbare have been obtained over the past few years, and now there are reference genomes representative for each of the 15 subpopulations [56] discovered through the 3K dataset. Genomic variation discovered from these reference genomes is expected to be more complete and more accurate since a closer reference genome has more similar sequence content in terms of population-specific SNPs, insertions/deletions as well as duplications and other copy-number variation (Figure 2a). It also becomes increasingly clear from pangenome studies that a linear reference genome is not a good enough data structure for representing the population data [71]. To be able to accurately call variants and represent in one dataset the variation coming from diverged lineages one needs a graph structure (Figure 2b). This is an active area of research, which has been already used in cattle and soybean [72,73]. This is also the topic of a new effort, Pan*Oryza*, led by Prof. Andy Jones, University of Liverpool, with collaborators from the European Bioinformatics Institute (EMBL-EBI), Oregon State University, University of Arizona, and IRRI.

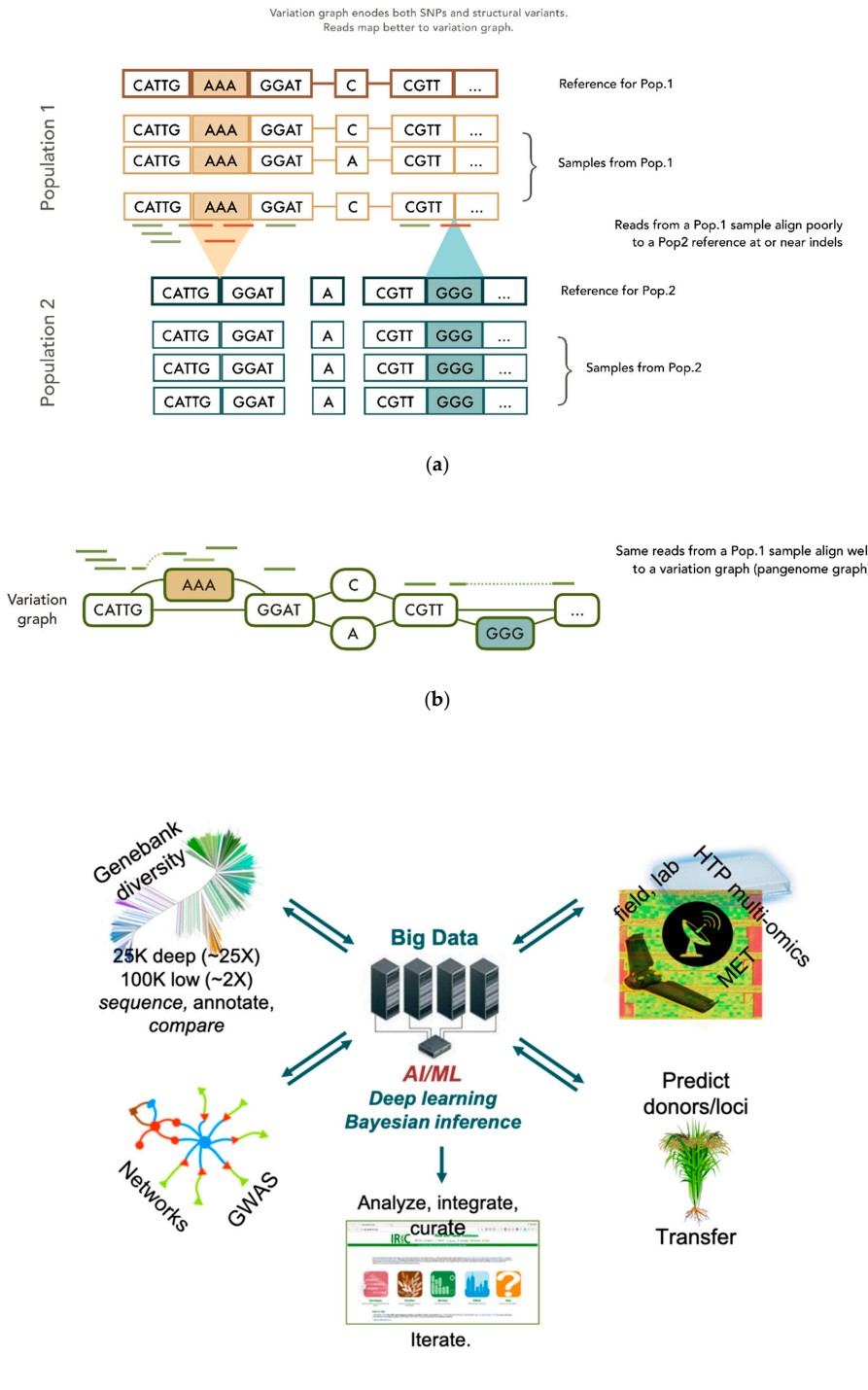

**Figure 2.** New strategy to enhance the genomic/genetic research using rice genetic resources. (**a**) Population specific references are necessary for better read mapping/variant discovery: without a reference that is from sample's population, the reads near indels are unmapped or poorly mapped; (**b**) A "variation graph" (VG or "pangenome graph") data structure represents all types of variants and allows for more accurate read mapping; (**c**) a schematic of digital rice genebank providing in-depth sequencing data for over 10,000 diverse rice accessions stored at the International Rice Genebank (IRG) and associated data, databases, tools, and analyses. AI/ML: artificial intelligence/machine learning, HTP: high throughput, MET: multi-environment testing, GWAS: genome-wide association study.

These progresses made in recent years brought on greater confidence to use genotype data in improving elite rice varieties. A digital rice genebank through collaboration between IRRI and partners in U.S., Saudi Arabia, and China aims to generate in-depth sequence data (at more than double the coverage depth obtained in the 3K dataset) using an additional set of over 10,000 rice accessions (Figure 2c) [74]. This expansion of sequencing analysis to a larger germplasm set possibly allows us to discover more subpopulations existing in nature, which leads to generate more reference genomes for future genetic studies. Furthermore, pan*Oryza* analysis may produce more accurate gene models that can replace simple gene models currently available in the rice annotation project databases. The greater depth and coverage of diverse germplasm in a digital genebank will enable the discovery of rare allelic variants that may boost the accuracy and efficiency in rice breeding to enhance desired traits in the near future. Having high coverage data on a significant fraction of the genebank will allow the remaining accessions to be sequenced at lower depths followed by imputation within the subpopulations of the genetic diversity to predict missing allelic states.

As of now, reference genomes for all AA genome species (*Oryza sativa, O. rufipogon, O. nivara, O. meridionalis, O. glaberrrima, O. barthii, O. longistaminata, O. glumaepatula*) and many other genome types are available [75]. Furthermore, work is under way to improve the quality and complete reference genomes for the remaining species. Population re-sequencing data (like 3K but smaller) are available for *Oryza glaberrima* and *O. rufipogon* (~400 accessions), and these resources will continue to grow. Comparative analyses across the genomic diversity will facilitate understanding of the evolution of the genus and its adaptation across different ecologies and under diverse conditions. Access to the high-quality reference builds across all of the wild *Oryza* will enable deeper exploration of these resources and discovery of novel genes and alleles that have been lost in domesticated rice and enable their use for improvement of cultivated rice.

A key requirement for success of large-scale sequencing projects is building an efficient data storage and processing infrastructure. Storing large sequence data in a traditional relational database management system (RDBMS) leads to inefficiency in data retrieval, as there is no natural support for fast regional and sample subset queries. Solutions to this can be multiple, from custom file formats storing matrix-shaped sequence data, to specialized formats like HDF5 or NetCDF holding array-based data, to advanced databases [76]. Currently, in rice we have the SNP-Seek database that uses a hybrid data storage architecture (variation data in HDF5 files, and phenotype and metadata in relational DB), with some forms of the data (raw BAM and VCF files) available pro bono through Amazon Web Services platform [77]. In addition, recently Google has released variant analysis from 3K project in public domain accessible through BigQuery database [78], allowing efficient access and advanced querying for power users. Solutions like this are expected to be used either alongside digital genebanks or as integral part of those, as the number of sequences increases, and may become indispensable for species with large genome size. For image data and derived phenotype data, there does not seem to be a standard solution yet, but examples tried in the community include CropSight [79], BreedBase, IAP (Integrated Analysis Platform) [80], utilizing both relational databases (MariaDB in CropSight) or NoSQL databases (such as MongoDB in IAP); see also PhenoImage [81]. Thus, we see a fully digitized genebank as a system comprising several components or modules, centered around a data storage and processing module, including phenotype data ingestion modules, and a rich web interface allowing users to interact with the data and mine genotype-phenotype associations using either built-in or external platform for statistical and ML-based analysis (Figure 2c).

## 5. Dissemination of Information for Better Use of Genetic Resources

As discussed above, the most critical requirement for the benefit-sharing of PGR is a user's access to information about the genetic resources held in the collections. To increase the accessibility, global crop communities have developed several public databases providing passport, phenotypic, and genotypic data of crop diversity stored at genebanks around the world. Genesys contains information on millions of accessions located in over 450 institutes [82]. It provides a quick-search function that allows users

to simultaneously see the information of hundreds of accessions with a single filter and then easily request seeds through the system. To integrate and reuse all of agronomic, socioeconomic, and digital data from the CGIAR system with partners, the BigData platform has been launched [83]. The Global Agricultural Research Data Innovation & Acceleration Network (GARDIAN), the CGIAR flagship data harvester provides a huge amount of information from more than 170,000 publications and 27,000 datasets [84]. To ensure the integrity and accessibility to various rice research data, IRRI has implemented several Big Data activities. These include developing databases that tag datasets with rice crop and other ontology descriptors, and AGROVOC and GACS (The Global Agricultural Concept Scheme) terms. AGROVOC is a controlled vocabulary covering all areas of interest of the Food and Agriculture Organization (FAO) of the United Nations while GACS (The Global Agricultural Concept Scheme) provides a hub for concepts related to agriculture, in multiple languages, for use in linked data. Other efforts are in place to create databases for UAV-based high throughput phenotypes and GIS applications [58,83,85,86].

## 6. Conclusions

As the impact of climate change seriously threatens crop productions worldwide, scientific innovations and advanced technologies promoting the use of PGR are in high demand. Although the genetic diversity held at genebanks contains useful genes and alleles to potentially address all kinds of constraints in crop production, it is difficult to select the most appropriate set of accessions without sufficient information such as phenotypic and genotypic data. Hence, researchers and breeders tend to use a small portion of the genetic diversity in their research programs, which indicates a big gap between genebank collections and breeding products. High throughput phenotyping performed during the routine operations of the genebanks can generate very useful information which helps the seed users to save resources for germplasm screening and further discovery steps. Access to a digital rice genebank, wherein all of the conserved accessions have genomic data, is now within reach, although it requires major support and dedicated teams. However, there are major constraints to be overcome, including the need for baseline, ground-truth phenotyping data to allow the prediction of phenotype from genotype and the development of new and improved tools enabling these predictions, such as those described by van Eeuwijk et al. [87]. Once in hand, these rich resources, when coupled with detailed multi-omics-including genomics proteomics, and metabolomics-on selected materials, will allow selection of genotypes and novel alleles and haplotypes for any trait. Exploration of these resources is expected to accelerate the breeding process while enhancing the use of the rich conserved diversity. The integrated databases launched by the CGIAR system provide comprehensive information on PGR and represent a first step to accelerating the use of genebanks accessions.

**Author Contributions:** J.-S.L., D.C., K.M., and A.K.; writing—original draft preparation, J.-S.L., D.C., J.D.P., K.M., and A.K.; writing—review and editing. All authors have read and agreed to the published version of the manuscript.

**Funding:** This research was funded by IRRI CRP Funding from the CG.

**Acknowledgments:** We thank Steve Klassen and Stephen Timple at IRRI for technical support.

**Conflicts of Interest:** The authors declare no conflict of interest.

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
