# Peer review of "Advanced Strategic Research to Promote the Use of Rice Genetic Resources"

_agronomy, doi:10.3390/agronomy10111629_

Round 1
Reviewer 1 Report
This manuscript focuses on specific issues of gene banks, specifically the high number of accessions contrasting with breeding needs and use –with emphasis on Rice, and the potential solution involving ‘new’ technologies such as automated phenotyping of seeds, drone assessment of plant status in the fields, genomics, and collective database of passport data for conserved accessions. The manuscript is generally well written (a few suggestions listed below though), though some paragraphs would need better transitions as they read as separate boxes and not always as coherent parts of a global text.
The manuscript unfortunately suffers from a strong weakness: none of issues at stake are really new --including suggested potential solutions, and justifications or arguments developed are naive at best, simply follow conventional wisdom, and drawbacks and alternative ways are not enough discussed. As a result, the reader is left wondering what he has learned from the text.
As a revised version with improved discussion and stronger case could possibly be of interest to a large audience.
General comment:
As highlighted above, a revised version should make a stronger case as to why the issues discussed can be solved with greater emphasis on technology and database. These issues are known since some time already, and new technologies do not seem to have actually solved any of the issues of abundance of accessions available for plant breeding. The authors should carefully make their case, but also discuss alternative ways, and clearly state their position with more compelling arguments (i.e., beyond the obvious). In the current version, their point seems to be discussing about impact of high throughput phenotyping approaches, yet none of the cited literature is actually dealing with the concept, either from a theoretical or from a factual standpoint. If my assessment is correct, than they need to stuff the work with more relevant references. If it is not, this means their point is not correctly conveyed at all and they strongly need to frame issues in a much better way. Also, the authors need to move beyond naive arguments, such as “advanced technologies and strategies developed through the global rice research programs will be applicable for other crop species as well”. They should realise this claim be true only for most closely related species, but certainly not for crops with very different life histories (e.g. vegetative reproduction), mating systems (e.g. obligate allogamous or self incompatible species), organ harvested (e.g. tuber), or gene bank status (e.g., seed recalcitrant species and trees).
Specific comments:
L42 to 47 : These sentences are a bit unclear (e.g. the use of “continuum” is confusing), please reformulate.
L58: please clarify how Rice could be a model for other crop species. It is unclear how a species with such a dramatic collect effort, full sequenced genomes available and long history of breeding schemes including interspecies introgression might help any research in unrelated and very different orphan crop species with much less if any sampling effort and no history of breeding schemes.
L63: genetic erosion can also occur by drift. Please reformulate.
L129: “are recommend” please reformulate.
L129: the full paragraph was rough in the flow, and the next paragraph is too. This is because they focus each on a specific evaluation need but are not linked to each other or the previous parts of the text. Please smooth reading for readers with introductory and final sentences that will ease reading flow.
L159: “are obligate” please replace with “have the moral obligation”
L255: “As the climate change impact” please delete “the”
L256: “demanded”. “required” would probably best fit intended meaning.
Author Response
This manuscript focuses on specific issues of gene banks, specifically the high number of accessions contrasting with breeding needs and use –with emphasis on Rice, and the potential solution involving ‘new’ technologies such as automated phenotyping of seeds, drone assessment of plant status in the fields, genomics, and collective database of passport data for conserved accessions. The manuscript is generally well written (a few suggestions listed below though), though some paragraphs would need better transitions as they read as separate boxes and not always as coherent parts of a global text.
The manuscript unfortunately suffers from a strong weakness: none of issues at stake are really new --including suggested potential solutions, and justifications or arguments developed are naive at best, simply follow conventional wisdom, and drawbacks and alternative ways are not enough discussed. As a result, the reader is left wondering what he has learned from the text.
As a revised version with improved discussion and stronger case could possibly be of interest to a large audience.
General comment:
As highlighted above, a revised version should make a stronger case as to why the issues discussed can be solved with greater emphasis on technology and database. These issues are known since some time already, and new technologies do not seem to have actually solved any of the issues of abundance of accessions available for plant breeding. The authors should carefully make their case, but also discuss alternative ways, and clearly state their position with more compelling arguments (i.e., beyond the obvious). In the current version, their point seems to be discussing about impact of high throughput phenotyping approaches, yet none of the cited literature is actually dealing with the concept, either from a theoretical or from a factual standpoint. If my assessment is correct, than they need to stuff the work with more relevant references. If it is not, this means their point is not correctly conveyed at all and they strongly need to frame issues in a much better way.
Response: we made more discussions and explanations why we believe the technologies we discussed have NOT been successful in the past but can be successful now. We have strengthened the aspect of high throughput phenotyping with adding two relevant references (line 158-170, 323-326). In the new paragraph (line 263-283), we discussed more practical approaches to facilitate digital gene bank activities, which describes a stronger case that IRRI and partner institutes are pursuing
Also, the authors need to move beyond naive arguments, such as “advanced technologies and strategies developed through the global rice research programs will be applicable for other crop species as well”. They should realise this claim be true only for most closely related species, but certainly not for crops with very different life histories (e.g. vegetative reproduction), mating systems (e.g. obligate allogamous or self incompatible species), organ harvested (e.g. tuber), or gene bank status (e.g., seed recalcitrant species and trees).
Response: we agree on this point and revised the sentences in the Abstract and Introduction accordingly
Specific comments:
L42 to 47 : These sentences are a bit unclear (e.g. the use of “continuum” is confusing), please reformulate.
Response: this section has been rewritten to improve clarity
L58: please clarify how Rice could be a model for other crop species. It is unclear how a species with such a dramatic collect effort, full sequenced genomes available and long history of breeding schemes including interspecies introgression might help any research in unrelated and very different orphan crop species with much less if any sampling effort and no history of breeding schemes.
Response: additional explanation was added to improve clarity
L63: genetic erosion can also occur by drift. Please reformulate.
Response: corrected
L129: “are recommend” please reformulate.
Response: corrected
L129: the full paragraph was rough in the flow, and the next paragraph is too. This is because they focus each on a specific evaluation need but are not linked to each other or the previous parts of the text. Please smooth reading for readers with introductory and final sentences that will ease reading flow.
Response: it is true that drone-phenotyping also contains high resolution image analysis which is a similar technology of the automated-phenotyping described in the previous paragraph. We have tried to smooth a reading flow by adding an additional paragraph integrating both field- and lab-based high throughput phenotyping
L159: “are obligate” please replace with “have the moral obligation”
Response: corrected
L255: “As the climate change impact” please delete “the”
Response: corrected
L256: “demanded”. “required” would probably best fit intended meaning.
Response: we replaced it with “in high demand”
Reviewer 2 Report
This manuscript adds to the literature on enhancing the value of plant genetic resources held in gene banks by applying various high-throughput technologies. It is not completely novel in that regard, but makes a contribution by focusing specifically on rice. The text is informative about the range of technologies available and the figures are helpful in illustrating some of the concepts. I think the ideas expressed are generally sound, but the authors tend to oversimplify the process of combining genomic and phenotypic data sets to identify key regions of the genome and then to pinpoint the accessions that meet a breeder’s requirements. Adding some discussion of the challenges involved, especially for traits that are highly variable in their expression, would provide a more realistic view of the state of the art.
The manuscript would benefit from a thorough review and editing by all the authors. There are grammatical errors, awkward phrasing, and parts that are difficult to follow. I’ve pointed out a few of these below, but a complete review would be a big benefit to the manuscript. Surely IRRI has people who are skilled at this type of thing.
Some specific editing suggestions follow.
Line 17 and elsewhere. ‘Drone’ is an ambiguous term. I think what the authors mean is unmanned aerial vehicle (UAV).
Line 21. Replace ‘and’ with ‘to’.
Line 35. ‘the year’ can be deleted.
Line 43. Add ‘of’ after ‘challenges’.
Line 44. Suggest ‘… critical proportions more quickly than current crop breeding cycles can respond to.’
Line 56-57. Other important reasons for sequencing rice were its relatively small genome size and close relationship to other major grain crops.
Line 66. Replace ‘prevent’ with ‘reduce’, as genetic erosion can also occur within gene banks.
Line 68. Add ‘ex situ’ before ‘collections’. ‘On the other hand’ does not fit here and can be deleted.
Line 75. Is qLG-9 a single locus or multiple loci as indicated?
Line 76. Add a comma after ‘longevity’ to improve comprehension.
Line 85. Replace ‘Contrastingly’ with ‘In contrast’.
Line 91. Add ‘for’ after ‘Pre-screening’.
Line 95. ‘routine operations’ may not be obvious to some readers. Suggest adding examples, ‘such as regeneration of seed or monitoring seed vigor’.
Line 102ff. Stay-green is most informative in a stress situations such as drought or heat. However, seed regeneration is not advisable under those conditions.
Line 116. ‘without cost’? Surely there is a cost involved in preparing the samples for imaging, collecting the data, and analyzing it.
Line 143-144. The meaning here about customized products allowing institutes to save their budgets is not clear.
Line 156-157. Figure 1 (c) caption. Indicate that accessions are planted in the same positions in both sides of the field (if that is the case).
Line 180-182. This makes the process sound too easy.
Line 200-201. ‘Reference genomes using three popular varieties … have been steadily accumulating’. Does this mean multiple genome sequences of the same three varieties?
Line 213. ‘over additional 10,000 rice accessions’. This needs rephrasing, maybe ‘an additional set of over 10,000 rice accessions’.
Line 218. I think ‘variances’ should be ‘variants’.
Figure 2. What is VG in part (b)? In the caption for (b), delete ‘(a)’. The caption for (c) is not descriptive of what is shown in the image. Also, there are many acronyms that are not defined.
Line 228. Rather than ‘value’, perhaps ‘requirement’. And ‘access’ instead of ‘accessibility’.
Line 238-243. This is a very long and muddled sentence that is hard to follow. Maybe it can be broken into 2 or 3 sentences.
Line 244. Instead of ‘AA genomes’, ‘AA genome species’. Some description of what these are would be helpful to non-rice researchers.
Line 252-253. Is the last sentence redundant with the previous one? If not, some further explanation of what is meant would be useful.
Line 256. ‘in high demand’ rather than ‘highly demanded’.
Line 266. ‘multi-omics’ should be further described.
Line 271. This section seems to end abruptly. A concluding sentence would be appropriate.
Line 273-274. The last sentence is duplicated.
Author Response
Comments and Suggestions for Authors
This manuscript adds to the literature on enhancing the value of plant genetic resources held in gene banks by applying various high-throughput technologies. It is not completely novel in that regard, but makes a contribution by focusing specifically on rice. The text is informative about the range of technologies available and the figures are helpful in illustrating some of the concepts. I think the ideas expressed are generally sound, but the authors tend to oversimplify the process of combining genomic and phenotypic data sets to identify key regions of the genome and then to pinpoint the accessions that meet a breeder’s requirements. Adding some discussion of the challenges involved, especially for traits that are highly variable in their expression, would provide a more realistic view of the state of the art.
Response: we carefully made revisions to address this valuable concern of the reviewer. In line 165-172, 254-284, 324-327 and elsewhere, we tried to explain the constraints in the use of rice genetic resources and possible solutions that rice research communities including IRRI are practicing. The truth is that we are still in the developing stage of impeccable solutions for maximizing the use of genetic resources, hence current manuscript may afford an opportunity for the global research community to keep improving the PGR research strategies
The manuscript would benefit from a thorough review and editing by all the authors. There are grammatical errors, awkward phrasing, and parts that are difficult to follow. I’ve pointed out a few of these below, but a complete review would be a big benefit to the manuscript. Surely IRRI has people who are skilled at this type of thing.
Response: the entire manuscript has been revised accordingly through a complete review
Some specific editing suggestions follow.
Line 17 and elsewhere. ‘Drone’ is an ambiguous term. I think what the authors mean is unmanned aerial vehicle (UAV).
Response: corrected
Line 21. Replace ‘and’ with ‘to’.
Response: corrected
Line 35. ‘the year’ can be deleted.
Response: corrected
Line 43. Add ‘of’ after ‘challenges’.
Response: corrected
Line 44. Suggest ‘… critical proportions more quickly than current crop breeding cycles can respond to.’
Response: corrected
Line 56-57. Other important reasons for sequencing rice were its relatively small genome size and close relationship to other major grain crops.
Response: corrected
Line 66. Replace ‘prevent’ with ‘reduce’, as genetic erosion can also occur within gene banks.
Response: corrected
Line 68. Add ‘ex situ’ before ‘collections’. ‘On the other hand’ does not fit here and can be deleted.
Response: corrected
Line 75. Is qLG-9 a single locus or multiple loci as indicated?
Response: corrected
Line 76. Add a comma after ‘longevity’ to improve comprehension.
Response: corrected
Line 85. Replace ‘Contrastingly’ with ‘In contrast’.
Response: corrected
Line 91. Add ‘for’ after ‘Pre-screening’.
Response: corrected
Line 95. ‘routine operations’ may not be obvious to some readers. Suggest adding examples, ‘such as regeneration of seed or monitoring seed vigor’.
Response: corrected
Line 102. Stay-green is most informative in a stress situations such as drought or heat. However, seed regeneration is not advisable under those conditions.
Response: it is true that screening for stay-green (SG) would be ideal under such stress conditions. Nevertheless, we would like to recommend gene bank managers to observe the SG during seed regeneration due to two reasons. Firstly, we have observed a big variation in SG at the regeneration field in both tropical and temperate regions and some rice accessions maintained dark-green leaf colour at the late seed maturity stage under normal weather conditions. Secondly, as we experienced during the last few years, climate change impact has increased the frequency of hot and humid weather. Even rice farmers in the temperate regions such as Korea and Japan experienced high rate of infertility due to sudden high temperature at flowering stage. This explanation is now added in the text
Line 116. ‘without cost’? Surely there is a cost involved in preparing the samples for imaging, collecting the data, and analyzing it.
Response: corrected
Line 143-144. The meaning here about customized products allowing institutes to save their budgets is not clear.
Response: we noticed this sentence is not necessary in the paragraph. It is now deleted
Line 156-157. Figure 1 (c) caption. Indicate that accessions are planted in the same positions in both sides of the field (if that is the case).
Response: this explanation is now added. Thank you
Line 180-182. This makes the process sound too easy.
Response: we added discussions about challenges/possible solution during this process
Line 200-201. ‘Reference genomes using three popular varieties… have been steadily accumulating’. Does this mean multiple genome sequences of the same three varieties?
Response: the text was edited for clarity
Line 213. ‘over additional 10,000 rice accessions’. This needs rephrasing, maybe ‘an additional set of over 10,000 rice accessions’.
Response: corrected
Line 218. I think ‘variances’ should be ‘variants’.
Response: corrected
Figure 2. What is VG in part (b)? In the caption for (b), delete ‘(a)’. The caption for (c) is not descriptive of what is shown in the image. Also, there are many acronyms that are not defined.
Response: corrected
Line 228. Rather than ‘value’, perhaps ‘requirement’. And ‘access’ instead of ‘accessibility’.
Response: corrected
Line 238-243. This is a very long and muddled sentence that is hard to follow. Maybe it can be broken into 2 or 3 sentences.
Response: corrected
Line 244. Instead of ‘AA genomes’, ‘AA genome species’. Some description of what these are would be helpful to non-rice researchers.
Response: we added a description
Line 252-253. Is the last sentence redundant with the previous one? If not, some further explanation of what is meant would be useful.
Response: this sentence is omitted
Line 256. ‘in high demand’ rather than ‘highly demanded’.
Response: corrected
Line 266. ‘multi-omics’ should be further described.
Response: examples of omics are added
Line 271. This section seems to end abruptly. A concluding sentence would be appropriate.
Response: for better flow of the full paragraph, we moved this section to the digital gene bank chapter
Line 273-274. The last sentence is duplicated.
Response: this sentence has been omitted